# The Development of Methods for the Production of New Molecular Vaccines and Appropriate RNA Fragments to Counteract Unwanted Genes: A Pilot Study

**DOI:** 10.3390/vaccines11071226

**Published:** 2023-07-11

**Authors:** Iskra Sainova, Vera Kolyovska, Iliana Ilieva, Tzvetanka Markova, Dimitrina Dimitrova-Dikanarova, Radka Hadjiolova

**Affiliations:** 1Institute of Experimental Morphology, Pathology and Anthropology with Museum (IEMPAM) to Bulgarian Academy of Sciences (BAS), 1113 Sofia, Bulgaria; 2Department of Pharmacology and Toxicology, Medical University of Sofia, 1431 Sofia, Bulgaria; 3Department of Biology, Medical University of Sofia, 1431 Sofia, Bulgaria; 4Department of Pathophysiology, Medical University of Sofia, 1431 Sofia, Bulgaria; radkakir@abv.bg

**Keywords:** in vitro incubated cells, adeno-associated virus DNA vectors, vaccine avipoxviral strains, intra- and extracellular virus forms

## Abstract

The potential of viruses as appropriate vectors for the development of new therapeutic strategies, as well as for the design of molecular (DNA, RNA, and/or protein) vaccines via substitution of nucleotide sequences, has been proven. Among the most appropriate DNA and/or RNA fragments, members belonging to families *Parvoviridae* (particularly adeno-associated virus, AAV) and *Poxviridae* have frequently been suggested for this purpose. In previous studies, the vaccine avipoxvirus strains FK (fowl) and Dessau (pigeon) have been proven able to infect mammalian cells (as well as avian cells), and to replicate productively in a small number of them; thus, we may be able to adapt them using incubation, and in these conditions. Additionally, we have previously proved, based on AAV recombinant DNA vectors, that it is possible to transfer appropriate genes of interest via mouse embryonic stem cells (mESCs). In the current study, we develop methods for the application of the same vaccine avipoxviral strains, based on the AAV DNA genome recombinant constructs, to be used for gene transfer in cells, for the transfer of DNA and/or RNA fragments (for the suppression of unwanted viral and/or cellular genes), and for the production of molecular (DNA, RNA, and/or protein) anti-cancer and anti-viral vaccines. To this end, sub-populations of embryonic mammalian cells infected with the two forms of both vaccine avipoxviral strains were frozen in the presence of cryo-protector dimethylsulfoxide (DMSO), subsequently thawed, and re-incubated. In most cases, the titers of the intra-cellular forms of the two strains were higher than those of their extra-cellular forms. These data were explained by the probable existence of the intra-cellular forms as different sub-forms, including those integrated in the cellular genome proviruses at a given stage of the cellular infection, and suggest the possibility of transferring nucleotide (DNA and/or RNA) fragments between cellular and viral genomes; this is due to the influence of activated fusion processes on DMSO, as well as drastic temperature variations.

## 1. Introduction

The potential of viruses as appropriate vectors for the development of various therapeutic strategies, but also for the design of molecular (DNA, RNA and/or protein) vaccines via the substitution of nucleotide sequences, has been proven [1]. In this context, viral vectors may be applied to the reparation of cellular genes, as well as of genes in other viruses that are related to viral infections. On the other hand, genes that cause infections and viruses that cause malignant cellular transformations, which are used as the basis of viral vector designs, should be eliminated.

When discussing the possibilities of utilizing viruses, those possessing RNA and/or DNA genomes [2,3] have been proven effective, alongside bacteria plasmids’ and yeasts’ genomes [4,5]. The initial vector constructs that are used should contain specific restriction sites, and the respective DNA fragment(s) of interest should be obtained using a restriction enzyme (in particular, bacterial endo-nucleases) treatment. Possible modifications could increase the expression level of the foreign gene via a change in the promoter, or in the insertion site (the place where the copy of the respective gene of interest is inserted) of the applied viral vector [6]. These features have also been taken into consideration in the design of different types of vaccines against coronaviruses [7].

The function of the cellular angiotensin-converting enzyme 2 (ACE2), as the main target of SARS-CoV-2/COVID-19, as well as of the other members belonging to the *Coronaviridae* family (the agents that cause SARS and MERS), has been proven [8,9]. Each monomer of the trimeric spike (S) glycoprotein consists of two subunits, S1 and S2, which bind to the ACE2 receptor, followed by the fusion of the viral and host cellular membranes [10]. Some differences in the arrangement of the envelope (E) and membrane (M) among SARS-CoV-2, SARS-CoV, and MERS-CoV have been shown [11,12]. Thus, together with the development of methods for the activation of adequate antiviral immunity, strategies for the suppression of key viral genes (those responsible for the virus’ penetration in the cell, those responsible for its replication, and those responsible for both processes) in the virus RNA genome are also necessary. To this end, the viral gene coding viral S protein should be suppressed with appropriate siRNAs. On the other hand, methods for designing molecular vaccines (with DNA, RNA and/or protein nature) that also counteract viral protein(s), such as E and M (but not the viral S protein), should be developed.

The early regions of the adenoviral (*Ad*) DNA genome (*Adenoviridae* family) have been determined with *E1*–*E4* transcripts (of the early genes), along with a full-length major late transcript, followed by the late *L1*–*L5* transcripts (of the late genes), produced by alternative splicing [13]. The inverted terminal regions (ITRs) serve as the origins of replication, and Ψ as a packaging sequence. When the early region *E1* in the Ad DNA vector is deleted, the inserted gene of interest normally replaces the deleted *E1* region, and its expression is driven by a heterologous promoter.

Among the most desirable and proven therapeutic carriers of DNA and RNA fragments is the adeno-associated virus (AAV) (of the *Parvoviridae* family), mainly because of the non-pathogenic nature of its wild-type and its ability to infect non-dividing terminally differentiated cells (as well as immature stem/progenitor cells), but also because of its sustained transgene expression in vivo [14,15,16,17]. Another appropriate and applicable substance for these goals has been characterized: the DNA genome of poxviruses (of the *Poxviridae* family) [18]. The expression of the inserted copy/copies of one or more gene(s) of interest could be achieved via integration of appropriate promoter gene, which should be located out of the *tk* locus of the gene, coding the enzyme thymidinkinase (TK) in the virus DNA genome, thanks to the proven necessity of intact *tk* gene. The production of recombinant vectors and vaccines based on the poxviral DNA genome have most often been achieved via homologous recombination or via direct molecular cloning [19]. Different methods and techniques for the maximally safe application of the received recombinant constructs on both cellular and organism levels should be developed; these methods depend on the type of the respective cells and organisms, as well as the type of the respective biological species. In this context, some of the necessary characteristics are application of attenuated vaccine viral strains, the appropriate initial infection titer (the concentration of infectious particles in a minimal volume), and the heterologous strains of the respective cells, tissues, organs and/or organisms; this is because, for instance, the application of heterologous avian viral and bacterial strains may not work with mammals and/or cells of mammalian origin [20]. The potential of avian pox viruses as appropriate vectors for the development of various therapeutic strategies with mammals and mammalian cells has been proven in several studies in the literature [21,22,23]. As it was proven in our previous studies that incubating and replicating the vaccine avian pox viral strains FK (fowl) and Dessau (pigeon) is possible in mammalian cells [24] besides in avian cells [25], this was taken in consideration. Similar results have also been observed by other authors, both in vitro and in vivo [23,26,27].

The main goal of the current study was to develop new methods for the application of viruses as vectors in the design of maximally safe new molecular anti-SARS-CoV-2/COVID-19 vaccines to counteract other virus proteins (i.e., those different from viral protein S), and to simultaneously boost the use of appropriate RNAs against unwanted virus genes (such as coding viral protein S); we aim to apply of these techniques to prevent the occurrence of other unwanted viral and/or cellular genes.

## 2. Materials and Methods

### 2.1. Cellular Cultures

Cells from the mammalian embryonic cell line EBTr, derived from embryonic bovine trachea cells [28], were incubated in initial volume 3 × 10^4^ in 1 mL cultural fluid, in a growth medium, presenting combination of Parker-E199 (Sigma, Kawasaki, Japan) and Iskov’s modification of Dulbecco’s medium (IMDM-Sigma), in ratio 1:1, supplemented with 25 mM HEPES buffer (Sigma), 5% normal bovine serum (NBS-Sigma) and/or 5% fetal bovine serum (FBS-Sigma), and antibiotics mixture (100 IU/mL Penicillin-Sigma and 100 μg/mL Streptomycin-Sigma). The incubation was performed in a humidified 5% CO_2_/95% air incubator at temperature 37 °C). The vitro-incubated cells were observed at each 24 h by inverted light microscope Televal, supplied by mega-pixel CCD-camera.

### 2.2. Previously Designed Recombinant DNA Viral Vectors, Based on Adeno-Associated Virus DNA Genome, Containing Inserted Nucleotide Genes/DNA Fragments

Commercial recombinant DNA constructs (Sigma-Aldrich, St. Louis, MO, USA), designed on the basis of adeno-associated virus (AAV) (*Parvoviridae* family) DNA genome, were applied. The tested genes of interest, inserted in each of the recombinant DNA vectors, were the oncogene *Dcn1* and the tumor-suppressor gene *HACE-1*, both isolated from mouse embryonic 3T3 fibroblasts from Balb/c mice. Aside from the respective gene of interest, the other obligatory components were the promoter of gene, coding eukaryotic elongation factor 1-alpha (EF1-α), isolated from 3T3 mouse embryonic fibroblasts, as well as isolated from bacterial DNA plasmid marker gene, providing resistance to Neomycin. The protein product of oncogene *Dcn1* is a protein-kinase, which is known to make key proteins of apoptosis easier targets for degradation [28,29,30,31,32]. This oncogene is conservative in all eukaryotic organisms, with respective analogue in prokaryotes, characterizing with high polymorphism in the different organisms. On the other hand, tumor-suppressor gene *HACE-1* codes protein-kinase which, in opposite to kinase protein product of oncogene *Dcn1*, makes easier targets for the degradation of key proteins of the malignancy and metastasis processes [30,31,32,33]. The design of the received recombinant DNA vector with the presence of the described genes was performed by treatment with appropriate ligases after previous treatment with specific restriction enzymes (bacterial endo-nucleases). Genomic assays to prove the presence of the respective gene(s) of interest were performed by electrophoresis on 1% agarose gel after standard polymerase chain reaction (PCR), and the expression of the same gene(s) after reverse transcriptase PCR (RT-PCR), respectively. Single-strand conformational polymorphism (SSCP) assay was also used for assessment of genetic variations, by application of the protocol developed by Dong and Zhu (2005) [34]. This method has also recently been applied in diagnostic and identification of mutations in new viral strains, including belonging to *Coronaviridae* family [35]. In all cases, specific 3′- and 5′-DNA primers (Sigma-Aldrich) were applied, complementary to the used recombinant DNA constructs (instead of the cellular DNA genome), and followed by electrophoresis in 1% agarose gel (Sigma-Aldrich) with previously added 1% solution of propidium iodide (Sigma-Aldrich). At the same time, the normal presence and expression of the tumor-suppressor gene *HACE-1* in normal cells was proved by electrophoresis in 1% agarose gel (Sigma-Aldrich) after performing of standard PCR and RT-PCR, respectively, of DNA material from normal mature epithelial cells of adult Balb/c experimental mice. In these cases, specific 3′- and 5′-DNA primers (Sigma-Aldrich) were applied, complementary to the cellular DNA. In all cases, the presence and size of the inserted copies of each respective gene of interest were determined on the basis of specific molecular DNA markers (Sigma-Aldrich) with size of the included DNA fragments from 750 to 2500 base pairs. The materials were provided by the Institute of Molecular Biotechnology (IMBA) to the Austrian Academy of Sciences in Vienna, Austria, by a short-term “Ernst Mach” fellowship/grant, where were performed the experiments described in this chapter.

### 2.3. Development of Methods for the Application of Vaccine Avipoxviral Strains as Vectors for the Design of Recombinant DNA Constructs, as Well as for the Exchange of Nucleotide (DNA and/or RNA Fragments) between Viral Particles and Cells

#### 2.3.1. Virus Inoculation of In Vitro-Incubated Cells

In vitro-incubated cultures of mammalian embryonic cells were inoculated in 24-well plates (24 Nunclon; Space Sever Flow Lab.; Linbro), as semi-confluent monolayers, with suspension of the avian DNA vaccine avian pox viral strains FK (fowl) and Dessau (pigeon) (*Poxviridae* family), with initial infectious titers 10^6.25^CCID_50_/mL and 10^5.0^CCID_50_/mL, respectively (which were the infectious doses, in which 50% pathological changes in cultures of primary embryonic duck cells, inoculated with each one of the two viral strains, were observed) [24]. The main goal of application of these comparatively low initial infectious titers (comparatively high initial dilutions of the viral suspensions) was related to the maximally safe application of the used vaccine avipoxviral strains to the in vitro-incubated mammalian cells. After absorption for 45 min at room temperature, all the inoculated cellular monolayers were washed three times with 1 mL in a well of phosphate-buffered solution (PBS, pH 7.2), and 1 mL of a well of supporting medium was added. Separate sub-populations of the same cells, inoculated with the two vaccine avipoxviral strains, were frozen in the presence of cryo-protector dimethylsulfoxide (DMSO), subsequently thawed, and re-incubated in fresh incubation medium at 37 °C in a humidified 5% CO_2_/95% air incubator. The results were received by light microscopy observations of the presence or absence of cytopathogenic effects in the in vitro-incubated cells. All virus-inoculated cell cultures were compared with the non-inoculated control cultures.

#### 2.3.2. Separation of Intra- and Extra-Cellular Forms of Each Vaccine Avipoxviral Strain on Each Cellular Type

After inoculation of all in vitro-incubated cell cultures described above, the intra-cellular forms of each one of the two vaccine avipoxviral strains were received by mechanical scraping of the virus-inoculated cellular monolayers. Extra-cellular forms of the same two virus strains were used as sources for the cultural fluids of the inoculated cells. De novo-formed semi-confluent monolayers of the same cells were inoculated with the two forms of each of the vaccine viral strains. The cells were inoculated with serial dilutions of the viral suspensions, and the respective infectious titer was determined by assessment of the dilution, in which, for the last time, the cytopathogenic effect on the cells (as a reciprocal value of the dilution) was observed.

#### 2.3.3. Statistical Analysis

The degree of the assessed cytopathogenic effect was compared between the separate virus-inoculated cell cultures and the times post viral inoculation. Subsequent software processing was performed. The values were expressed as mean ± standard deviation (SD), and a Student’s *t*-test was applied. The differences were considered as statistically significant in *p* < 0.01.

## 3. Results

### 3.1. Investigation on the Presence and Expression of Inserted Copies of One or More Specific Genes of Interest in Previously Designed Recombinant DNA Constructs, Based on the AAV DNA Genome

The presence and expression of the inserted copy of the oncogene *Dcn1* (Figure 1A) and of the tumor-suppressor gene *HACE-1* (Figure 1B) was proven by application of standard PCR and RT-PCR of both “donor” (Figure 1A,B-*Up*) and “recipient” (Figure 1A,B-*Down*) recombinant DNA vectors. Specific primers, complementary to the respective recombinant DNA constructs (“donor” and “recipient”), were applied in the application of both standard PCR and RT-PCR. The normal presence and expression of the tumor-suppressor gene *HACE-1* in normal cells was confirmed by a standard PCR and RT-PCR of DNA material, isolated form normal mature epithelial cells of adult experimental mice Balb/c, by application of primers, complementary to the cellular DNA (Figure 1A).

### 3.2. Investigation of the Possibility for Productive and Non-Productive Replication of Vaccine Avipoxvirus Strains and Isolation of Intra- and Extra-Cellular Forms in Non-Permissive Mammalian Cells

In all cases (from the 24th to the 168th hours post viral inoculation of the in vitro-incubated mammalian embryonic cells), the titers of the extra-cellular forms of the two vaccine avipoxviral strains were significantly higher (with 0.5–1 logarithms) (Figure 2A) than their intra-cellular forms (Figure 2B). These results suggested productive viral replication with production of mature virions, which was accepted as a proof of a good adaptation of the tested viral strain on the respective cellular system.

The opposite tendency was noted when those inoculated with the same forms of the two vaccine avipoxviral strains mammalian embryonic cells were frozen in the presence of cryo-protector DMSO, subsequently thawed, and re-incubated (Figure 3). In all cases, the titers of the intra-cellular forms of both vaccine strains were significantly higher (Figure 3A) (with approximately 0.5–2 logarithms) than these of their extra-cellular forms (Figure 3B). These data suggested the eventual presence of the viral strains strain in various sub-forms in the cell, including as pro-viruses. These features were detected at a later stage in the two non-permissive avian viral strains mammalian cells (from the 72nd to the 168th hour post viral infection).

In most of the cases, the titers of both intra- and extra-cellular forms of the vaccine fowl pox viral strain FK were significantly higher than those of the vaccine pigeon pox viral strain Dessau (Figure 2 and Figure 3). These differences could be explained with the lower initial infectious titers of the pigeon vaccine viral strain than that of the vaccine fowl pox virus strain.

## 4. Discussion

The described previously designed recombinant gene constructs, based on the AAV DNA genome, have been applied in our previous studies about transfer of additional copies of oncogene *Dcn1* in in vitro-incubated mouse embryonic stem cells (mESC), and of additional copies of tumor-suppressor gene *HACE-1* in in vitro-incubated malignant human cervical carcinoma cells HeLa, respectively [36,37]. The decrease in the malignant potential of the cervical carcinoma cells by an additionally inserted copy of the tumor-suppressor gene and its protein product should be activated by mechanisms which also allow eventual degenerative changes to be escaped, by appropriate inter-molecular interactions [33]. On the other hand, the anti-aging functions of the oncogenes, and oncoproteins coded by them, should be activated, by which eventual infections, malignant cellular transformations, metastatic processes, etc., will be escaped; for instance, by generation of an adequate anti-malignant immune response on the influence of the oncoproteins in their role of active antigens [38]. Messages about the analogical possibility of generation and support of adequate immune responses by human trophoblasts, immortalized by inoculation with virus strain SV40, have been obtained [39] by activation of in the same cascade regulatory mechanisms [40].

The design of recombinant DNA constructs, containing copies of one or more gene(s) of interest, as well as for detection of ways for safe application, is necessary to develop novel recombinant (molecular) vaccines, but also for the needs of biotechnology, gene and tissue engineering, as well as for gene and cellular therapeutic procedures [1]. In penetration into the cell through the cellular receptor ACE2, the negative influence of the viruses, belonging to *Coronaviridae* family, but also of their separate components, on the normal functions of this cellular enzyme should be taken in consideration as underlining many symptoms of the infections caused by them, including injured functions of many important anatomic organs [41,42]. Additionally, the role of the renin–angiotensin–aldosterone system (RAAS) has been proven as key to the regulation of systemic blood pressure and renal functions [43]. Thus, methods and strategies for improving and regulation of the functions of cellular and viral proteins should be developed [44].

Vectors, based on the genomes of adeno-associated viruses and poxviruses, have shown some advantages compared to vectors, based on the adenovirus DNA genome, containing non-coding DNA to achieve enough genome stability [45,46,47]. When an early region in such a vector is deleted, the inserted gene of interest normally replaces the deletion of a late region and its expression is driven by a heterologous promoter. In this way, most of the vectors based on the adenovirus genome have been characterized with removed or deleted late regions but, because such regions are not essential for the viral replication, the possibility for insertion of an ~8 kb foreign expression cassette has been proved.

The established higher titers of the extra-cellular forms of both vaccine avipoxviral strains, compared to their intra-cellular forms, proposed productive viral replication with the production of mature virions and, thus, the possibility for good adaptation of the tested viral strains on both cellular types [21,22,23,24,25,26]. On the other hand, the observed higher titers of the intra-cellular forms of the two vaccine avipoxviral strains than the titers of their extra-cellular forms suggested the existence of intra-cellular viruses as different sub-forms in the different stages of the cellular infection, including as pro-viruses, integrated into the cellular genome at any stage of the cellular inoculation [27,48]. The appearance of these features at a late time period in the non-permissive about avian viral strains mammalian cells (from the 72nd to the 168th hours post viral infection) were also in confirmation of the literature data [49]. Additionally, as a whole, the titers of the two forms of the pigeon strain were significantly lower compared to those of the fowl strain, which could be explained with the applied significantly lower initial infectious titers of the poxviral strain Dessau than those of the pox viral strain FK [24,28]. The noted changes, in particular in inoculation of the in vitro-incubated embryonic mammalian cells with the intra-cellular forms of the two vaccine avian pox viral strains, could be explained by the eventually activated fusion processes between cells and viruses, by the influence of cryo-protector DMSO, or by the drastic temperature changes. According to the scientific literature, a lot of properties of many membrane molecules influence DMSO [50,51,52] and of other organic detergents [53,54], as do drastic temperature changes [55], which could be changed. Many inter-molecular interactions have also proven to be influenced, which has also been proposed as a factor, activating the fusion processes [56]. In this way, the possibility for transfer of nucleotide (DNA and/or RNA) fragments from a virus to a cellular genome, as well as in the opposite direction (from cellular to viral genome) in these conditions could be proposed, and new methods for the production of novel safe recombinant DNA constructs, molecular vaccines (DNA, RNA and/or protein), as well as for appropriate siRNAs on the basis of adeno-associated virus/parvovirus and poxvirus genomes by appropriate application and processing, could be suggested [25,26,27].

## 5. Conclusions

The presence and expression of additionally inserted copies of the respective genes of interest in both “donor” and “recipient” recombinant constructs/vectors, based on the AAV DNA genome, were proven. The lower titers of both forms of vaccine strain Dessau on each of the two cellular types in most of the cases, compared to the titers of the same forms of the vaccine strain FK, could be explained with the significantly lower initial infectious titers of the pigeon strain than these of fowl strain. When the titers of the extra-cellular forms of both avipoxviral strains were higher compared with their intra-cellular forms, productive production of novel mature virions was proposed and, thus, good adaptation of the respective virus strain for incubation in the respective cellular system. On the other hand, in freezing the mammalian cells inoculated with the each one of the two vaccine strains in the presence of cryo-protector DMSO, subsequent thawing and re-incubation, higher titers of intra-cellular forms of both vaccine avipoxviral strains were observed compared with the titers of their extra-cellular forms, which suggested that the eventual existence of the intra-cellular forms as different sub-forms, including as proviruses integrated into the cellular genome in any stage of the cellular infection. These features with the two forms of vaccine avipoxviral strains were established at a late time period (from the 72nd to the 168th hours post viral inoculation), and could be explained with the activated fusion processes on the influence of DMSO and the drastic temperature changes. In this way, the current results suggested the possibility for successful application of AAVs (*Parvoviridae* family) and vaccine avipoxviral strains (*Poxviridae* family) for application as vectors for the design of recombinant DNA constructs, siRNAs, molecular (DNA, RNA and/or protein) vaccines, as well as to be used for reparation of viral and cellular genes by transfer of nucleotide (DNA and/or RNA) fragments between cells and viral particles.

## Figures and Tables

**Figure 1 vaccines-11-01226-f001:**
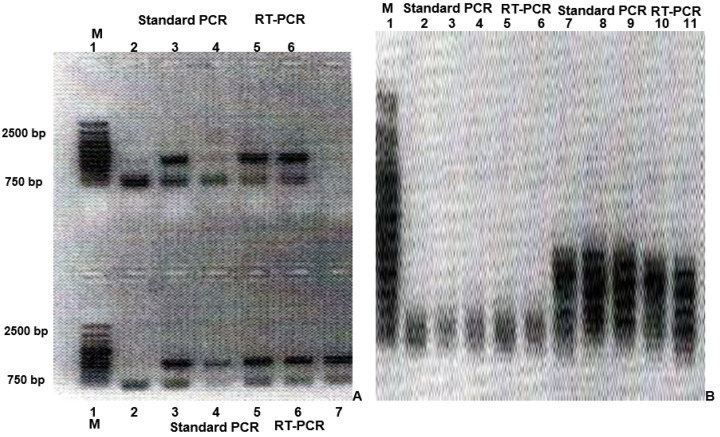
A 1% agarose gel electrophoresis to prove the presence of inserted copies of oncogene *Dcn1* and tumor-suppressor gene *HACE1* (**A**) of oncogene *Dcn1* by standard 1% agarose gel electrophoresis, and (**B**) of oncogene *Dcn1* and tumor-suppressor gene *HACE1* by electrophoresis after previous denaturation (singe-strand conformational polymorphism/SSCP assay). (**A**): *Up*—lane 1—molecular marker (M) of 750–2500 base pairs (bp); lanes 3 and 5 prove the presence of the inserted copy of oncogene *Dcn1* by standard PCR and specific primers, complementary to the used recombinant DNA construct; lane 6 proves the expression of the inserted copy of tumor-suppressor gene *Dcn1* by application of RT-PCR; *Down*—lane 1—molecular marker (M) of 750–2500 base pairs (bp); lanes 3 and 5 prove the presence of the inserted copy of oncogene *Dcn1* by standard PCR and primers, complementary to the used recombinant DNA construct; lanes 6 and 7 prove the expression of the inserted copy of oncogene *Dcn1* by RT-PCR and primers, complementary to the used recombinant DNA vector. (**B**): Lane 1—molecular marker (M); lanes 2–4 prove the presence of the inserted copy of tumor-suppressor gene *HACE-1* by standard PCR and primers, complementary to the used recombinant DNA construct; lanes 5 and 6 prove the expression of the inserted copy of tumor-suppressor gene *HACE-1* by RT-PCR and primers, complementary to the used recombinant DNA vector; lanes 7 and 9 prove the presence of the inserted copy of oncogene *Dcn1* by standard PCR and primers, complementary to the used recombinant DNA vector; lanes 10 and 11 prove the expression of the inserted copy of oncogene *Dcn1* by RT-PCR, and primers, complementary to the used recombinant DNA construct.

**Figure 2 vaccines-11-01226-f002:**
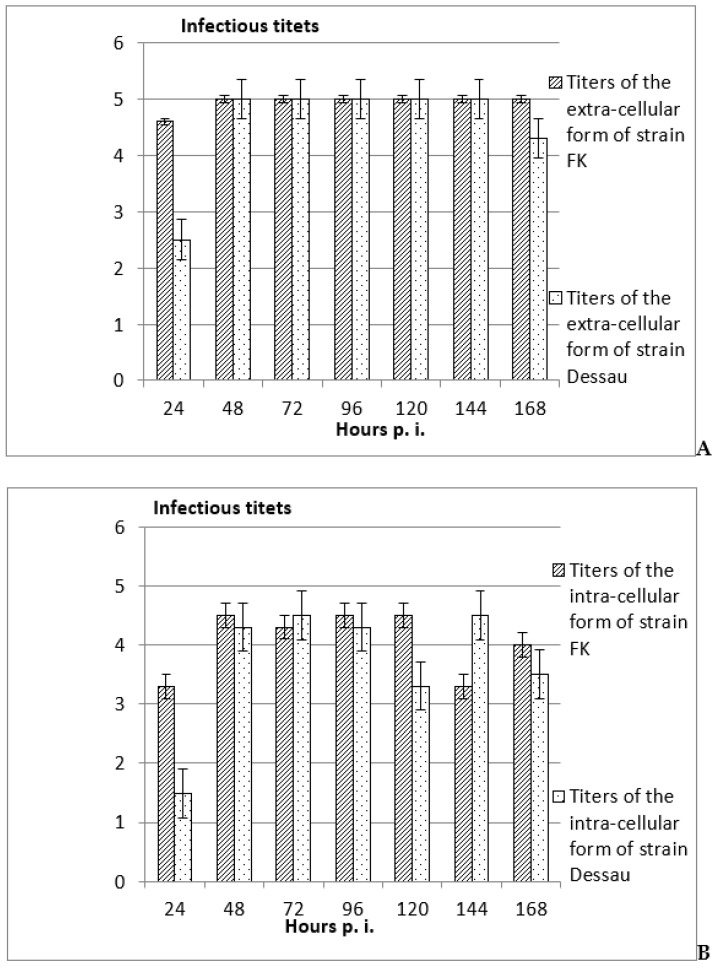
Titers of the extra-cellular (**A**) and intra-cellular (**B**) forms of the vaccine avipoxviral strains FK and Dessau, in heterologous of mammalian embryonic cells after adaptation of both strains, for productive replication in the cells from the same type, determined by assessment of the differences when comparing the respective values in the non-inoculated controls and subsequent software processing, ±SD, *p <* 0.01.

**Figure 3 vaccines-11-01226-f003:**
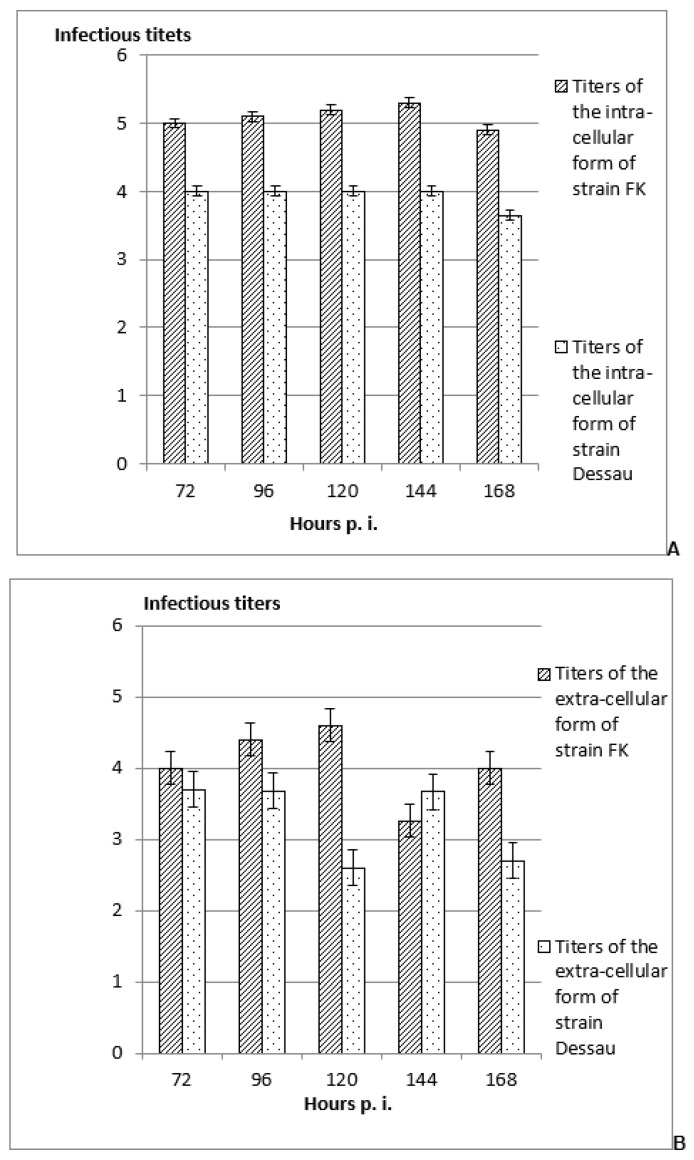
Titers of the intra-cellular (**A**) and extra-cellular (**B**) forms of the vaccine avipoxviral strains FK and Dessau, incubated in mammalian embryonic cells from EBTr cell line, followed by freezing in the presence of cryo-protector DMSO, subsequent thawing, and re-incubation. The data were determined by assessment the differences, comparing the respective values in the non-inoculated controls and subsequent software processing, ±SD, *p <* 0.01.

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
