# Peer review of "The Development of Methods for the Production of New Molecular Vaccines and Appropriate RNA Fragments to Counteract Unwanted Genes: A Pilot Study"

_vaccines, 2023, doi:10.3390/vaccines11071226_

Round 1

Reviewer 1 Report (Previous Reviewer 1)

Acceptable. 

Author Response

Reviewer 2 Report (New Reviewer)

This manuscript presented the development of a method of viruses as vectors for the design of new vaccines. However, the manuscript seems to be not ready as a formal submission. For example:

(1) a lot of text is in red;

(2) Figure 1 is very messy: the labeled numbers are not corresonded to the lanes;  "standard  PCR" and "RT-PCR", which are cut into two lines, look very confusing.

(3) Figure 2: "titets" should be "titers"; the legends are very crowded.

Other issues:

(1) The type of statistics analysis, e.g., one-tailed t-test, should be presented in Section 2.3.3;

(2) why no error bars fof the strain FK titers in Figure 3A?

Subjects are missing in many sentences.

For example, in the abstract:

"Among the most appropriate careers of DNA- and/or RNA-fragments have been suggested the members, belonging to families Parvoviridae (particularly Adeno-Associated Virus - AAV) and Poxviridae."

"In our previous studies has been proved a possibil- ity of vaccine avipoxvirus strains FK (fowl) and Dessau (pigeon) to infect besides avian cells, also mammalian cells and to replicate productively in a small number of them, and thus, to be adapted about incubation and replication in non-permissive about them conditions."

In introduction:

"Among the most desirable therapeutic carriers of DNA- and RNA-fragments has been proved Adeno-Associated Virus (AAV) (Parvoviridae family), mainly because of the non-pathogenic nature of its wild-type and its ability to infect besides unmature stem/pro- genitor cells, also non-dividing terminally differentiated cells, but also because of its sustained transgene expression in vivo [14-17]"

Round 2

Reviewer 2 Report (New Reviewer)

The issues are addressed in this revision.

This manuscript is a resubmission of an earlier submission. The following is a list of the peer review reports and author responses from that submission.

Round 1

Reviewer 1 Report

·       Abstract is poorly written. The authors have directly started the abstract with methods and results. Write few sentences about the objective/need of this study. The length of abstract should be reduced; currently it has 475 words.

·       There are grammatical mistakes throughout the manuscript. Some of the sentences are difficult to comprehend. Furthermore, the manuscript lacks coherence. The authors are advised to revise the manuscript to correct these flaws.

·       Abstract: “As sources of extra-cellular forms of the two viral strains served the cultural fluids, and of their intra-cellular forms – the scraped cellular monolayers”. What are the authors implying?

·       Introduction: ………..substitution of appropriate cell nucleotide sequences has been proved. What do the authors mean by cell nucleotide sequence?

·       “Possible modifications could be increased expression level of the foreign antigen by a change in the promoter or in the insertion site of the applied viral vector.” What does he author mean here “or in the insertion site of the applied viral vector”?

·       “Thus, together with development of methods for activation of adequate antiviral immunity, strategies for suppression of key viral genes (as genes, responsible about the virus penetration in the cell, for the virus replication, but also the two processes)”. ……..but also two processes???

·       “On the other hand, methods about development of molecular vaccines (with DNA-, RNA- and/or protein nature) against any other viral protein(s), different of viral S protein, should be developed.” …?????

·       “Its expression could be provided by integration of appropriate promoter gene, which should be located………..”. Expression of what?

·       “In this direction, the potential of avian pox viruses as appropriate vectors for development of various therapeutic strategies with mammals and mammalian cells has been          proved in many literature findings”. The authors say in many literature findings but have not cited any papers here.

·       “Proved in our previous studies possibility about incubation and replication of the vaccine avian pox viral strains FK (fowl) and Dessau (pigeon) besides in avian cells [20], also in            mammalian cells has been proved in our previous studies…”. The sentence begins and ends with “proved in our previous studies”.

·       What are non-wished virus genes?

            Materials and methods

o   Section 2.2  The obligatory components of the developed recombinant DNA-construct, designed on the basis of adeno-associated virus (AAV) (Parvoviridae family) DNA-genome, were the respective one or more gene(s) of interest, but also promoter of gene, coding eukaryotic elongation factor 1-alpha (EF1-α), isolated from 3T3 mouse embryonic fibroblasts from Balb/c laboratory mice, as well as isolated from bacterial DNA-plasmid marker gene, providing resistance to Neomycin”. What are the authors trying to convey??

o   Method used for the development of recombinant DNA viral vectors is not clear. What were the genes of interest and from what organism they were obtained?

o   Provide the list of primers used for PCR and RT-PCR studies.

            Results

·       “The presence and expression of the inserted copy of the respective gene of interest in the “donor” recombinant DNA-vectors…….”. What were the gene of interest? Their names are             not mentioned anywhere.

·       Figure 1 is difficult to understand. The figure is not labelled properly. Which is “A” and which is “B”. Rewrite the Figure 1 description. Label the gene of interest with the band size. The lane number and the wells in the agarose gels do not align.

·       Divide the results sections into different sub-sections? The authors have simply written all the outcomes of this study in one paragraph.

·       How did the authors measure the infectious titers of the intracellular and extracellular    apoxviral strains in different cell lines? It is not mentioned in the methods or results section? What statistical paramerters were used?  If there are microscopic images showing the infection of cell line by the virus strains provide them.

·       In figures 2 and 3 correct the spelling: “ infectious titets” to “infectious titers”.

            Discussion

·       “Methods and strategies for improving and regulation of the functions of cellular and viral proteins should be developed [30]. In penetration in the cell through the cellular re-ceptor ACE2, the negative influence of the viruses, belonging to Coronaviridae family, but even of their separate components, on the normal functions of this cellular enzyme should be taken in consideration as underlining in many symptoms of the infections, caused by them, including injured functions of many important anatomic organs [31,32]. Addition-ally, the role of renin–angiotensin–aldosterone system (RAAS) has been proved as key in the regulation of the systemic blood pressure and renal function [33]”. What is the purpose of these sentences? They are incoherent.

·       “Vectors, based on the genomes of adeno-associated viruses and poxviruses have shown some advantages compare with vectors….”. What advantages? Mention the advantages.

Conclusion

·       The authors have simply rewritten the sentences from results and discussion in the conclusion. Rewrite the conclusion. Mention future possibilities and possible limitations of this study. 

Author Response

My co-authors and me have read the recommendation of reviewer #1, we have made the recommended corrections in the current research paper, and now we would wish to send as attached PDF-file Responses to reviewer #1 point-by-point

Reviewer 2 Report

The author doesn't identify the gene that was inserted into the adenoviral vectors.  In figure 1, the molecular weight markers are not visible in the gel.  What size of DNA was inserted?  What gene was inserted? With no molecular weight markers, the size of fragments are difficult to evaluate and verify. 

In figures 2 and 3, the data should be presented as box and whisker plots.  The significant values and not identified within the figures 2 and 3.  In the text, Figure 1A is written on page 5, is this supposed to be Figure 2A? The growth rate of the recombinant viruses should be compared on the different cell cultures against the growth rate of the parent adenovirus from fowl (Fk) along with comparing the recombinant virus from Fk to that recombinant virus from pigeon (Dessau).

What are the "non-wished virus genes"?

Author Response

We have written the recommendation, made by reviewer #2, we have made the recommended corrections in the current research paper, and now we would wish to send our replies point-by-point as attached PDF-file

Round 2

Reviewer 1 Report

Authors have done most of the editing. However, I request to check for typos and removing redundant sentences, especially in Introduction and discussion sections. 

Author Response

Additional clarifications and descriptions in the "Abstract", "Introduction"; "Materials and Methods"; "Results"; "Discussion" and "Conclusion" were made, related mainly with the tested genes of interest, which were inserted in the commercial recombinant gene constructs, based on the AAV DNA-genome, and the respective literature data in this connection were added and cited, but on the other hand, the methods of derivation of intra- and extra-cellular forms of comparatively low initial infectious tites (comparatively high initial dilutions of the viral suspensions, respectively) of partially attenuated by many passages vaccine strains of heterologous about mammals and mammalian cells avian viral strains (for investigation of possibilities about their application as vectors for design of molecular DNA-, RNA- andor protein vaccines, as well as of other materials for gene-engineering goals) 

Reviewer 2 Report

The authors revised the manuscript to mention the genes that were inserted; however, do not state the size of the genes that were inserted into the Avian Adenovirus vectors.  The figure 1 has not been revised.  The molecular weight markers are not clear.  It is impossible to determine the size of the fragments amplified since the molecular weight marker is not defined and fragment sizes are not identified.  There is no molecular weight marker on the figure 1B.  In addition, statistical significant values are not identified in figures 2 and 3.  The FK and Dessau derived AAV will grow at different rates and titers in the different avian cell lines, more importantly, how does the insertion of the different genes Dcn1 and/or HACE1 affect growth rate and titers compared to the parent AAV without the inserted genes?

Author Response

The provided to us recombinant gene constructs, based on the AAV DNA-genome, each one containing inserted copy of the respective tested gene of interest, were commercial and the copies of one and of the other gene of interest were previously inserted in them. Commercial were also the used DNA-markers and primers (3' and 5'), as well as the matrials for performing of standard PCR and of reverse transcriptase PCR (RT-PCR), including for preparation of agarose gels. In this relation, I have added information about "Ernst-March" grant in helping the performance of these experiments.

The poxviruses are separate part, they have no parent relationship with the used above AAV recombinant DNA-constructs. In this part, we make an initial step in development of methods about application of Poxvirus DNA-genomes as vectors about design of molecular DNA-, RNA- and/or protein vaccines, as well as of other materials for gene-engineering and genes reparation goals (as a new species besides AAV and Adenovirus DNA genomes and Retrovirus RNA-genomes). So, in the current study we tested comparatively low initial infectious titers (comparatively high initial dilutions of the viral suspensions, respectively) of partially attenuated by many passages vaccine strains of heterologous about mammals and mammalian cells avian pox viral species.

Some revisions in "Figure 1" were made, including information about the sizes of the used marker was added. Some explanations and clarifications were added about the statistical assay, mainly in "Materials and Methods" and "Results". The used avian pox virus strains FK and Dessau have no relationship with the used AAV vector - the investigation on previously prepared recombinant vectors on the basis of DNA-genome were tested for exchange to gene/DNA-fragments (copies), and the possibility about future application of poxviruses about the same goals was also tested and methods for this goal were developed (as application of comparatively low initial infectious titers/high initial dilutions of the viral suspensions of partially attenuated by many passages vaccine strains of heterologous about mammals and mammalian cells avian pox viral strains were, for provide of their maximally safe application).
